# Overexpression of an Antioxidant Enzyme APX1 in *cpr5* Mutant Restores its Pleiotropic Growth Phenotype

**DOI:** 10.3390/antiox12020301

**Published:** 2023-01-28

**Authors:** Fan Qi, Jianwei Li, Xiufang Hong, Zhiyi Jia, Binyan Wu, Fucheng Lin, Yan Liang

**Affiliations:** State Key Laboratory for Managing Biotic and Chemical Threats to the Quality and Safety of Agro-Products, Institute of Biotechnology, Zhejiang University, Hangzhou 310058, China

**Keywords:** ascorbate peroxidase, CPR5, leaf senescence, reactive oxygen species, salicylic acid

## Abstract

Breeding crops with enhanced immunity is an effective strategy to reduce yield loss caused by pathogens. The *constitutive expresser of pathogenesis-related genes* (*cpr5*) mutant shows enhanced pathogen resistance but retarded growth; thus, it restricts the application of *cpr5* in breeding crops with disease resistance. Reactive oxygen species (ROS) play important roles in plant growth and defense. In this study, we determined that the *cpr5* mutant exhibited excessive ROS accumulation. However, the mutation of respiratory burst oxidase homolog D (RBOHD), a reduced nicotinamide adenine dinucleotide phosphate (NADPH) oxidase responsible for the production of ROS signaling in plant immunity, did not suppress excessive ROS levels in *cpr5*. Furthermore, the *cpr5* mutant showed low levels of ascorbate peroxidase 1 (APX1), an important cytosolic ROS-scavenging enzyme. *APX1* overexpression in the *cpr5* background removed excessive ROS and restored the pleiotropic growth phenotype. Notably, *APX1* overexpression did not reduce the resistance of *cpr5* mutant to virulent strain *Pseudomonas syringae* pv. *tomato* (*Pst*) DC3000 and avirulent strain *Pst* DC3000 (*avrRpt2*). These results suggest that the removal of excessive ROS by *APX1* overexpression restored the *cpr5* growth phenotype while conserving pathogen resistance. Hence, our study provides a theoretical and empirical basis for utilizing CPR5 in the breeding of crops with disease resistance by effective oxidative stress management via APX1 expression.

## 1. Introduction

The *constitutive expresser of pathogenesis-related genes* 5 (*cpr5*) was initially identified by screening for mutants with enhanced disease resistance [1]. The CPR5 protein has the N-terminal conserved serine/arginine-rich RNA recognition motif (RRM) domain and multiple transmembrane domains at the C-terminus [2], acting as an element in RNA processing complexes in nuclear speckles and a component of the pore complex in nuclear membranes [2,3]. The *cpr5* mutants were also isolated by many other types of mutant screenings [4,5,6]. Therefore, *cpr5* mutants have pleiotropic growth phenotypes such as decreased trichome branching, premature leaf senescence, and reduced stomatal aperture, even under normal growth conditions [1,7,8]. Phytohormone signaling pathways are major regulators of plant growth and stress response [9]. *cpr5* has been used to elucidate the link between plant development and phytohormone signaling [10,11]. However, none of the hormone-deficient mutants suppressed the pleiotropic growth phenotype, especially premature leaf senescence in the *cpr5* mutants. 

Reactive oxygen species (ROS) are molecules containing active oxygen. They include superoxide, the hydroxyl radical, singlet oxygen, and hydrogen peroxide (H_2_O_2_) [12]. ROS play important roles in plant growth, development, and biotic and abiotic stress responses [13,14]. However, excessive ROS cause oxidative stresses, DNA and membrane protein damage, and eventual cell death [15]. Many disease-resistant plant mutants display high basal ROS levels causing lesions, mimicking cell death, retarding growth, and reducing yield [16,17]. It has been suggested that excessive ROS are the major reason for the pleiotropic phenotype in *cpr5* [11]; however, it is unknown whether growth and pathogen resistance in *cpr5* can be uncoupled by removing excessive ROS.

ROS are generated during photosynthesis in chloroplasts, respiration in mitochondria, and photorespiration in peroxisomes [18]. They are also produced by NADPH oxidases, which are known as respiratory burst oxidase homologs (RBOHs) in plants [19]. As ROS are highly reactive, their levels must be controlled to avoid cytotoxicity. Plants have evolved sophisticated non-enzymatic and enzymatic ROS-scavenging antioxidant defense systems in all their cell compartments [18,20]. The non-enzymatic systems are mediated mainly by ascorbate-glutathione [21]. The enzymatic systems include ascorbate peroxidases (APXs), catalases (CATs), and glutathione peroxidases (GPXs) [21]. Cytosolic APX1 is essential for the protection of chloroplasts from oxidative stress [22]. Loss-of-function *apx1* mutants exhibit enhanced sensitivity to various abiotic stressors such as high temperature, high light intensity, salt, drought, and heavy metals [22,23,24,25]. Recently, studies determined that APX1 could catalyze luminol-based chemiluminescence assays and allow monitoring of cytosolic ROS accumulation. Loss-of-function of APX1 results in cytosolic ROS accumulation but reduced light signals in the luminol–H_2_O_2_–APX1 reaction upon treatment with avirulent bacteria and lipopolysaccharide (LPS), an elicitor from the outer membrane of Gram-negative bacteria [26].

In this study, we established that the previously found *delt9* (*defective in LPS-triggered ROS production*) mutation [27,28] was a novel allele of the *CPR5* gene. *delt9* and *cpr5* showed reduced light signals in the luminol–H_2_O_2_–APX1 reaction but enhanced ROS levels according to colorimetric and fluorescent dye staining. The introduction of *rbohD* into the *cpr5* mutants did not suppress ROS accumulation. In contrast, *APX1* overexpression restored ROS levels in *cpr5*. We also determined that *APX1* overexpression restored the pleiotropic growth phenotype but not pathogen resistance in the *cpr5* mutants. Overall, our results suggest that *cpr5*-mediated disease resistance could be uncoupled from plant growth impairment by removing excessive ROS via *APX1* overexpression.

## 2. Materials and Methods

### 2.1. Plant Materials and Growth Conditions

The genetic background of the wild-type and mutant *Arabidopsis thaliana* used in the present study was the Columbia-0 (Col-0) ecotype. Seeds of the *cpr5* (salk_074631), *rbohD* (CS9555), and *adr1-L2* (salk_126422) T-DNA insertion mutant were obtained from the Non-Profit Arabidopsis Share Center (https://www.arashare.cn). *APX::APX-GFP* was used as previously described [26]. The primers used to identify the homozygous lines are listed in Appendix A.

The seeds were surface-sterilized by immersion in 10% (*v*/*v*) sodium hypochlorite (NaOCl) for 10 min, rinsed thrice with sterilized water, and sown on half-strength Murashige and Skoog (1/2 MS) agar containing 0.5% (*w*/*v*) sucrose. The seedlings were raised in a growth chamber (Model No. GDN-260C-4; Ningbo Southeast Instrument Co., Ltd., Ningbo, China) at 22 °C, 14 h light/10 h dark photoperiod, 75% RH, and 15,000 lx. For the mature plant assays, 7-day-old seedlings were transplanted in soil (Sun Gro Horticulture:Hawita Professional = 1:1) and raised in a growth chamber (Model No. DFY-1000E-3; Ningbo Southeast Instrument Co., Ltd.) at 22 °C, 14 h light/10 h dark photoperiod, and 15,000 lx.

### 2.2. Next-Generation Sequencing (NGS)-Based DELT9 Cloning

The *delt9* mutant was backcrossed with Col-0, and F_1_ was self-pollinated to generate F_2_. Two pools with or without luminescent signals after LPS treatment were selected and subjected to whole-genome sequencing (WGS) on the Illumina X-ten System (Gene Denovo Biotechnology, Guangzhou, China). A genome-wide single-nucleotide polymorphisms (SNP) analysis detected the genomic region with relatively high SNP index and harboring a candidate SNP in *CPR5*.

### 2.3. Plasmid Construction and Transgenic Plant Generation

The primers used for gene cloning and plasmid construction are listed in Appendix A. All constructs were generated by the Gateway Cloning System (Thermo Fisher Scientific, Waltham, MA, USA). To generate the *35S::CPR5-HA* construct, cDNA sequences without *CPR5* stop codons were amplified by PCR, cloned into pDONR/Zeo plasmids via BP (gateway) cloning, and subcloned into pGWB14 vectors by LR (gateway) reaction. The constructs were then electroporated into *Agrobacterium tumefaciens* GV3101. Transgenic plants were obtained using the *Agrobacterium*-mediated floral dipping transformation method [29].

### 2.4. RNA Isolation and qRT-PCR

Total foliar RNA was extracted with TRIzol reagent (Thermo Fisher Scientific). The cDNA was synthesized using 1 μg total RNA and a HiScript III 1st-Strand cDNA Synthesis Kit (Vazyme Biotech, Nanjing, China). ChamQTM SYBR^®^ qPCR Master Mix (Vazyme Biotech) was used for the qRT-PCR, and *ACTIN7* was the reference gene. Relative gene expression levels were calculated by the 2^−ΔΔCt^ method [30]. All primers used in the qRT-PCR are listed in Appendix A.

### 2.5. Protein Extraction and Immunoblot Analysis

Protein extraction and immunoblot analysis were performed as previously described [26]. APX1 proteins were extracted with buffer consisting of 50 mM Tris-HCl (pH 8.0), 150 mM NaCl, 1% (*v/v*) Triton X-100, and 1% (*v*/*v*) protease inhibitor cocktail (Merck KGaA, Darmstadt, Germany). For CPR5 and RBOHD protein detection, the extraction buffer consisted of 50 mM Tris-HCl (pH 5.7), 2% (*w*/*v*) sodium dodecyl sulfate (SDS), 1 mM phenylmethylsulfonyl fluoride (PMSF), and 1% (*v*/*v*) protease inhibitor. The tissues were ground, resuspended in the protein extraction buffer, and incubated on ice for 30 min. Cell debris was removed by centrifugation at 12,000× *g* and 4 °C for 15 min. The extracted proteins were separated by 8% SDS-PAGE (for CPR5 and RBOHD) and by 12% SDS-PAGE (for APX1) and transferred to polyvinylidene difluoride (PVDF) membranes at 80 V and 4 °C for 3 h. The membranes were blocked with 5% (*w*/*v*) nonfat milk and incubated with horseradish peroxidase (HRP)-conjugated α-HA (Roche Diagnostics, Basel, Switzerland), α-RBOHD (Agrisera, Vännäs, Sweden), α-cAPX (Agrisera), and α-ACTIN (ABclonal, Wuhan, China) antibodies. Signals were detected with a SuperSignal West Femto Trial Kit (Thermo Fisher Scientific).

### 2.6. Measurement of APX Activity

APX activity was measured as previously described [26]. Briefly, total proteins were extracted from four-week-old leaves using extraction buffer containing 50 mM pH 7.8 phosphate-buffered saline (PBS) and 200 μM ethylenediaminetetraacetic acid (EDTA). Protein concentration was determined by a Bradford assay [31]. APX activity was measured by monitoring the decrease in absorbance at 290 nm for 2 min after adding the supernatant, 5 mM ascorbate, and 20 mM H_2_O_2_.

### 2.7. ROS Detection by Luminol-Based Chemiluminescence Assay

ROS signals were measured by luminol-based chemiluminescence assay according to a previously described procedure [27]. Leaf disks (0.2 cm^2^) from 4-week-old plants were incubated overnight in water with light exposure in a 96-well plate. Then, 50 μL of a solution consisting of 200 mM luminol and 50 mg/mL LPS (L9143; Merck KGaA) was added to each well, and the chemiluminescence signals were recorded with a Photek HRPCS5 camera (HRPCS5; Photek, East Sussex, UK).

### 2.8. ROS Detection by 3,3′-Diaminobenzidine (DAB) Staining

DAB staining was performed as described previously with some modifications [19]. Excised leaves were vacuum-infiltrated with DAB staining solution (pH 6.0) consisting of 1 mg/mL DAB, 10 mM Na_2_HPO_4_, and 0.05% (*v*/*v*) Tween 20 and incubated in the dark at 24 °C for 6–8 h until brown precipitate was observed in the leaves. The leaves were then decolorized with de-staining solution ( ethanol:acetic acid:glycerol = 3:1:1 (*v*/*v*/*v*)). Leaves that remained dark brown contained H_2_O_2_ and were viewed under a light microscope (Nikon, Tokyo, Japan). The relative DAB staining intensity per unit leaf area was quantified, and the H_2_O_2_ content was determined using the ImageJ software (National Institutes of Health [NIH], Bethesda, MD; https://imagej.nih.gov/ij/; accessed on 19 March 2022.).

### 2.9. ROS Detection by Fluorescent 2,7-Dichlorodihydrofluorescein Diacetate (H_2_DCFDA) Staining

The H_2_DCFDA staining assay was performed according to a previously reported method with slight modifications [32]. The excised leaves were stained with 10 μM H_2_DCFDA (MedChemExpress, Monmouth Junction, NJ, USA) in 10 mM PBS buffer in the dark for 30 min. Images were captured under an Olympus FV3000 confocal laser scanning microscope (Olympus Corp., Tokyo, Japan) with a 488 nm filter. ROS signals were visualized in the range of 501–550 nm, and chlorophyll autofluorescence was detected in the range of 640–735 nm.

### 2.10. Mitochondrial Superoxide Detection by MitoSOX Red Staining

Mitochondrial superoxide was imaged by MitoSOX Red staining [33]. MitoSOX Red (RM02822, ABclonal) selectively targets the mitochondria, is oxidized by the superoxide there and becomes fluorescent. Lower leaf epidermis was stripped from leaf segments with a razor blade; the stripped leaf segments were then floated on 5 μM MitoSOX Red in 10 mM KH_2_PO_4_ alkalized to pH 7.4 with KOH in the dark at 24 °C for 30 min. The leaf segments were then rinsed thrice in the dark with 10 mM KH_2_PO_4_ alkalized to pH 7.4 with KOH. The samples were then mounted on microscope slides with the stripped (abaxial) leaf surfaces facing the coverslip. The mesophyll cell layers were immediately examined with an Olympus FV3000 confocal laser scanning microscope at 488/585–615 nm excitation/detection.

### 2.11. H_2_O_2_ Quantification by Titanium Sulfate Assay

The H_2_O_2_ was measured by titanium sulfate assay as previously described with slight modifications [34]. Briefly, 0.1 g fresh leaves were frozen in liquid nitrogen and ground. Each sample was suspended in 0.5 mL cold acetone and centrifuged at 8000× *g* and 4 °C for 15 min. Then 0.1 mL of 5% (*w*/*v*) Ti(SO_4_)_2_ and 0.2 mL NH_4_OH were added to 0.4 mL of the supernatant. The suspension was centrifuged at 5000× *g* for 15 min, and the precipitate was collected. Then 1 mL acetone was added to it, and the mixture was centrifuged at 5000× *g* for 15 min until a white precipitate formed. The precipitate was then dissolved in 1 mL of 2 M H_2_SO_4,_ and absorbance was read at 415 nm. The output was converted to μmol H_2_O_2_/g leaf tissue.

### 2.12. Trichome Imaging

The trichomes on living leaves were photographed with a Nikon Digital Sight DS-Fi2 camera mounted on a Nikon SMZ18 microscope.

### 2.13. Bacterial Growth Assay

Leaves of 4-week-old Arabidopsis plants were infiltrated with *Pseudomonas syringae* pv. *tomato* (*Pst*) DC3000 or *Pst* DC3000 (*avrRpt2*) at 1 × 10^5^ CFU/mL. Then, 2 or 3 d after infiltration, three 1.5-cm^2^ leaf disks were ground in 500 mL of 10 mM MgCl_2_, and the suspension was serially diluted and plated on NYG medium consisting of 5 g/L peptone, 3 g/L yeast extract, 2% (*w*/*v*) glycerol, 1.5% (*w*/*v*) agar, and rifampicin. The colonies were enumerated 2 d after plating.

## 3. Results

### 3.1. DELT9 Encodes CPR5

LPS-triggered ROS bursts were monitored by the luminol-based chemiluminescence method in the cytosol, where APX1 catalyzed the luminol–H_2_O_2_ reaction [26]. Mutants *defective in LPS-triggered ROS production* (*delt*) were previously isolated from an ethyl methanesulfonate (EMS)-mutagenized Arabidopsis seedling population [27,28]. The *delt9* mutant exhibited lower luminescence signals than the wild-type (Col-0) after the LPS treatment (Figure 1A). 

We identified the *delt9* mutation by whole-genome sequencing-based cloning using an F_2_ population derived from the backcross between *delt9* and Col-0 (Appendix A). Three nonsynonymous SNPs were identified in the linkage region (Appendix A). Among these, a substitution of C to T at nucleotide 842 was found in At5G64930, leading to a Val-281 substitution for Ala-281 (A281V) in CPR5 protein (Figure 1B). To determine whether the *CPR5* mutation was responsible for the reduced luminescent signals in *delt9*, we expressed *CPR5* tagged with *HA* under the control of the cauliflower mosaic virus (CaMV) *35S* promoter in the *delt9* background (*35S::CPR5-HA*/*delt9*). We measured the CPR5-HA protein levels in the transgenic lines by immunoblotting with an α-HA antibody (Figure 1C). Subsequent measurement of the luminescent signals in both independent transgenic lines revealed that *CPR5* overexpression restored luminescence in *delt9* (Figure 1D). In addition, we obtained a T-DNA insertion line with the insertion site at the end of the 4th exon and designated it *cpr5* (Figure 1B), which displayed ~50% transcript reduction (Figure 1E and Appendix A). The *cpr5* mutants also displayed significantly decreased luminescence signals after LPS treatment (Figure 1F). Collectively, these results indicate that it was the mutation in *CPR5* that reduced luminescence during the luminol–H_2_O_2_–APX1 reaction.

### 3.2. delt9 and cpr5 Mutants Accumulate Excessive ROS

The *delt9* and *cpr5* mutants showed reduced luminol–H_2_O_2_–APX1 signals, possibly because of low H_2_O_2_ levels. We tested this hypothesis by staining the leaves of the wild-type and those of the *delt9* and *cpr5* mutants with DAB. H_2_O_2_ oxidizes DAB, and the reaction product is a dark brown precipitate that colors the leaves [35]. We observed strong dark brown color in the leaves, and the relative intensity was *delt9* > *cpr5* > wild-type (Figure 2A,B). These results suggest that the *delt9* and *cpr5* mutants might have elevated H_2_O_2_ concentrations. The total H_2_O_2_ in leaves was extracted and spectrophotometrically quantified by the titanium sulfate–H_2_O_2_ reaction, which forms a yellow precipitate [36]. The *delt9* and *cpr5* mutants had significantly higher H_2_O_2_ levels than the wild-type (Figure 2C). We also examined intracellular ROS distribution by H_2_DCFDA staining. This probe exhibits green fluorescence after it is oxidized [37]. Stronger green fluorescence was observed in the chloroplasts of the *delt9* and *cpr5* mutants than in those of the wild-type (Figure 2D). However, as chloroplast fluorescence was extremely intense after H_2_DCFDA staining, it was difficult to observe ROS production in other organelles, for example, in mitochondria that produce superoxide and are closely associated with leaf senescence. Leaves of *delt9* and *cpr5* mutants were stained with MitoSOX Red, a mitochondrial ROS indicator that fluoresces red upon oxidation by superoxide [33]. We found that the mutants displayed far more intense red fluorescence than the wild-type (Figure 2E). Taken together, these results suggest that the *cpr5* mutants accumulated excessive chloroplastic and mitochondrial ROS.

### 3.3. Autoimmunity Is not Responsible for Premature Leaf Senescence in cpr5 Mutants

In addition to the intracellular location, ROS are produced in the apoplast by RBOHD in plant immunity [38]. Apoplastic ROS are difficult to detect by colorimetry or fluorescent dyes. Hence, we detected RBOHD protein abundance in the *delt9* and *cpr5* mutants via immunoblotting with an α-RBOHD antibody and determined that it was considerably higher in the *delt9* and *cpr5* mutants than in the wild-type (Figure 3A). We generated *cpr5 rbohD* double mutants to investigate whether an increase in RBOHD abundance contributes to excessive ROS in *cpr5*. The introduction of *rbohD* into the *cpr5* background did not reduce excessive ROS in this mutant according to DAB staining (Figure 3B,C). The 4-week-old *cpr5* mutants presented with premature leaf senescence. Their cotyledons were small and yellow, and their rosettes were curled in an abaxial direction (Figure 3D). The *cpr5 rbohD* double mutants displayed the same degree of premature leaf senescence as the *cpr5* single mutants. Together, the findings suggest that RBOHD does not explain excessive ROS production and premature leaf senescence in *cpr5*.

Recently, it has been reported that ACTIVATED DISEASE RESISTANCE 1 (ADR1), a helper nucleotide-binding leucine-rich repeat protein (NLR), is upregulated in double mutants with loss-of-function of BRASSINOSTEROID INSENSITIVE 1 (BRI1)-ASSOCIATED RECEPTOR KINASE 1 (BAK1) and its closest paralog BAK1-LIKE 1 (BKK1) [39]. The premature leaf senescence phenotype of *bak1 bkk1* is suppressed by ADR1 mutation. Given that high levels of *ADR1s* were also found in the *cpr5* mutants in a previous study [40], we examined whether *adr1* can suppress the premature leaf senescence of *cpr5*. We found that the introduction of *adr1-L2* into *cpr5* did not restore the premature leaf senescence in *cpr5* (Appendix A), suggesting that premature leaf senescence of *cpr5* is not due to autoimmune responses mediated by ADR1.

### 3.4. cpr5 Mutants Displayed Reduced APX1 Abundance

The preceding results indicate that *cpr5* generated reduced luminol–H_2_O_2_–APX1 signals but elevated cytosolic H_2_O_2_ levels. Hence, *cpr5* might have low endogenous APX1 levels. We tested this hypothesis by measuring APX1 protein abundance in the *cpr5* mutants via immunoblotting with an α-cAPX antibody. Arabidopsis contains three cytosolic APX isoforms, including APX1, APX2, and APX6. *APX1* exhibits the highest expression under normal growth conditions and accounts for ~50% of soluble APX activity [41]. A null allele of *apx1* (named *delt4*) [26] was used as a negative control for the detection of APX1 using the α-cAPX antibody (Figure 4A). A band with the predicted size of APX1 (28 kD) was found in the wild-type but absent in the *delt4* mutant, suggesting that this α-cAPX antibody can be used to detect APX1 proteins (Figure 4A). The APX1 protein abundance was lower in *delt9* and *cpr5* than in the wild-type (Figure 4A). We then measured the soluble APX activity using ascorbate as a substrate. We found that the APX activity was significantly lower in *delt9* and *cpr5* than in the wild-type (Figure 4B). The introduction of *rbohD* into the *cpr5* mutants did not recover APX activity (Appendix A). To determine whether reduced luminol–H_2_O_2_–APX1 signals can be rescued by overexpressing *APX1*, we generated *pAPX1::APX1-GFP*/*cpr5* transgenic plants by crossing *pAPX1::APX1-GFP*/Col-0 with the *cpr5* mutants (Figure 4C). The *pAPX1::APX1-GFP* transgene restored APX activity (Figure 4D) and luminol–H_2_O_2_–APX1 signals in the *cpr5* mutants (Figure 4E,F). Collectively, these results indicate that the *cpr5* mutants had lower APX1 levels than the wild-type.

### 3.5. APX1 Overexpression in cpr5 Mutants Removes Excessive ROS 

APX1 can scavenge cytosolic H_2_O_2_; hence, we measure the H_2_O_2_ levels in *pAPX1::APX1-GFP*/*cpr5* transgenic plants using the DAB staining method. The brown precipitate in the *cpr5* mutants disappeared after *APX1* was introduced into the *cpr5* background (Figure 5A,B). We then measured the expression levels of several transcripts associated with ROS. *WRKY25*, *WRKY53*, and *WRYK75*, important transcription factors downstream of ROS signaling, were significantly downregulated in *pAPX1::APX1-GFP*/*cpr5* plants compared with those in the *cpr5* mutants (Figure 5C). *APX1* overexpression in the *cpr5* mutants downregulated antioxidant genes, including *MDAR3* encoding monodehydroascorbate reductase, *GSTFs* encoding Φ-class glutathione S-transferases, and *AOX1a* and *AOX1d* encoding mitochondrial alternative oxidase (Figure 5C). Moreover, *SENESCENCE-ASSOCIATED GENE 13* (*SAG13*) was significantly downregulated in *pAPX1::APX1-GFP*/*cpr5* compared to that in *cpr5* (Figure 5C). Collectively, these results demonstrate that *APX1* removes excessive cytosolic ROS and downregulates downstream ROS regulatory genes.

### 3.6. APX1 Overexpression in cpr5 Mutants Suppresses Pleiotropic Growth Phenotype, but Not Pathogen Resistance

Next, we investigated whether cytosolic ROS removal by *APX1* overexpression rescues the pleotropic growth phenotype in the *cpr5* mutants. First, the *pAPX1::APX1-GFP* expression in *cpr5* restored its premature leaf senescence (Figure 6A). Second, *pAPX1::APX1-GFP*/*cpr5* displayed normal trichome development with three branches, whereas *cpr5* showed mostly two branches (Figure 6B,C). Third, the reduced stomatal aperture in *cpr5* mutants was restored by the transgene of *pAPX1::APX1-GFP* (Figure 6D,E). These results suggest that *APX1* overexpression reversed the aforementioned abnormalities of *cpr5* growth. The *cpr5* mutants have been reported to display enhanced resistance to the hemibiotrophic pathogen *Pst* DC3000 [4]. We found that *APX1* overexpression in the *cpr5* mutants did not show significant differences from that in the *cpr5* mutants in resistance to *Pst* DC3000 (Figure 6F). We next examined plant resistance to an avirulent strain *Pst* DC3000 (*AvrRpt2*). The amount of *Pst* DC3000 (*AvrRpt2*) was significantly reduced in *pAPX1::APX1-GFP*/*cpr5* transgenic plants compared with *cpr5* mutants, suggesting that *APX1* overexpression further enhanced *cpr5* resistance to *Pst* DC3000 (*AvrRpt2*). Notably, *APX1* overexpression in the wild-type also enhanced its resistance to *Pst* DC3000 (*AvrRpt2*) (Figure 6G).Taken together, these results suggest that *APX1* overexpression in the *cpr5* mutants suppresses the pleiotropic growth phenotype without reducing pathogen resistance.

## 4. Discussion

Breeding crops with enhanced disease resistance is an effective yield loss reduction strategy; however, fitness costs are often associated with increased pathogen resistance [42,43]. Several approaches have been used to lower the fitness costs related to augmented host defense [43,44]. For example, defense responses may be induced only when they are absolutely required [45,46]. In the present study, we found that APX1 overexpression in the *cpr5* mutants restored growth impairment while maintaining enhanced disease resistance in Arabidopsis. Hence, we propose that the trade-off between growth and defense in *cpr5* mutants can be mitigated by regulating cytosolic ROS levels (Figure 7).

The balance between ROS generation and scavenging is vital for the maintenance of cellular ROS homeostasis [18]. ROS levels exceeding the antioxidant capacity of a system may result in oxidative stress [15]. Enzymatic and non-enzymatic antioxidant systems scavenge ROS and control oxidative stress signaling. Here, we found that *APX1* overexpression completely restored the growth phenotype and leaf senescence of the *cpr5* mutants. However, APX1 downregulation is unlikely to be the main reason for premature leaf senescence in *cpr5*, as the *apx1* mutants did not present with an aging phenotype as severe as that of *cpr5* [47]. Furthermore, a single mutation of other APX isoforms did not result in the severe premature senescence phenotype [47]. These observations suggest that other ROS-scavenging enzymes, such as CATs and GPXs, may also be inhibited and cause lesion formation and early senescence in *cpr5*.

ROS are by-products of cellular metabolism. In both normal and stressed plants, ROS are produced in the apoplast, chloroplasts, mitochondria, and peroxisomes [18]. The *cpr5* mutants are in a chronic state of oxidative stress [11]. However, the sources of excessive ROS accumulation in it remain largely unknown. Staining the *cpr5* mutants with H_2_DCFDA and MitoSOX Red disclosed that excessive ROS accumulated in their chloroplasts and mitochondria. Mitochondria-derived ROS trigger cell death in animals and plants [48]. Arabidopsis *mosaic death 1* (*mod1*) mutants presented with excessive ROS accumulation and premature leaf senescence [49]. Suppressor screening of *mod1* indicates that most mutants in which excessive ROS and cell death were suppressed were deficient in mitochondrial complex 1, suggesting that mitochondrial ROS are essential to trigger cell death [50]. *MOD1*, encoding an enoyl-acyl carrier protein reductase that participates in fatty acid biosynthesis in chloroplasts [49], triggers mitochondrial ROS accumulation via the malate shuttle [51]. NADPH oxidase is not responsible for *mod1* cell death, similar to its function for *cpr5* cell death [50]. Compared with the wild-type, the *cpr5* mutants contained higher AOX1a and AOX1d transcript levels. These enzymes prevent excessive ROS formation in the mitochondria [52,53]. These findings suggest that excessive mitochondrial ROS might contribute to the pleiotropic growth phenotype of *cpr5*. Furthermore, chloroplastic ROS induced by MAPK cascade activation can induce cell death [54]. The MAPK cascade mutants *mekk1* and *mpk4* accumulated abundant ROS and presented with an extremely dwarfed and premature leaf senescence phenotype resembling that of *cpr5* [55]. It is, therefore, possible that both chloroplastic and mitochondrial ROS are critical in the cell death of *cpr5*. Nevertheless, cytosolic APX1 overexpression can remove excessive ROS in *cpr5*. This discovery is consistent with the previous notion that APX1 is the master regulator of ROS homeostasis in intracellular organelles [22].

The *cpr5* mutants showed increased *Pst* DC3000 resistance mainly due to high levels of salicylic acid (SA), an important phytohormone in plant pathogen resistance [1,4,56]. SA degradation upon *nahG* introduction in *cpr5* restored its defenses to the wild-type levels, whereas the premature leaf senescence in *cpr5* was not fully recovered [11], suggesting that SA degradation might not remove the excessive ROS in *cpr5*. It is generally accepted that SA and ROS are mutually regulated and form a self-amplifying loop under biotic stress [57,58]. However, SA-regulated ROS is mediated by RBOHs, as SA-induced ROS production is absent in the *rbohD* mutants [27,59,60]. Consistently, we determined that the introduction of *rbohD* into *cpr5* neither removed its excessive ROS nor restored premature leaf senescence. These results suggest that high SA levels might not be the main cause of excessive ROS accumulation in *cpr5*, which might originate from intracellular organelles. Thus, this provides a possibility to uncouple ROS-mediated growth retardation from SA-mediated defense. 

## 5. Conclusions

Plants have evolved complex mechanisms to balance growth and defense in the process of adapting to diverse biotic and abiotic stressors. The *cpr5* mutants showed enhanced pathogen resistance but retarded growth. In the present study, we uncoupled plant growth from plant pathogen resistance by removing excessive ROS via *APX1* overexpression.

## Figures and Tables

**Figure 1 antioxidants-12-00301-f001:**
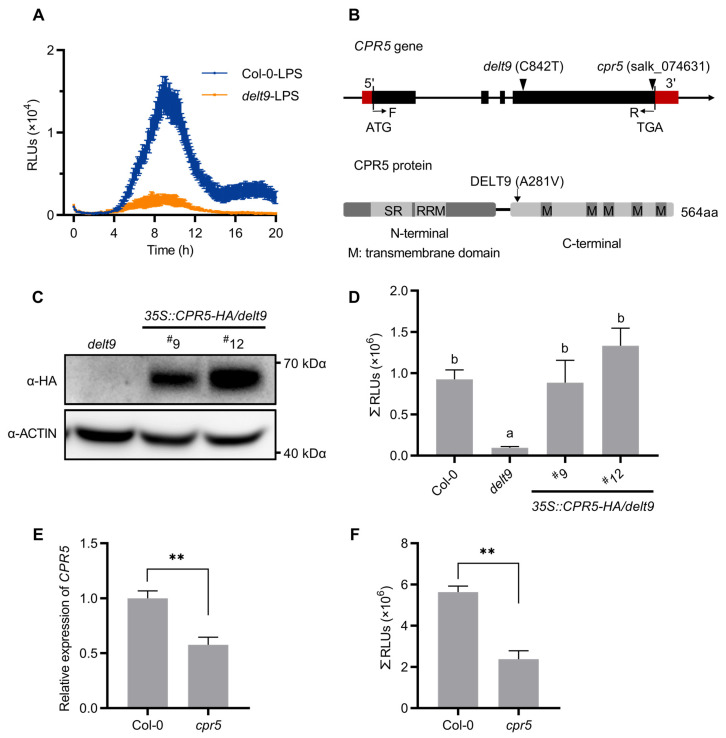
*cpr5* mutants show reduced luminol-emitted light signals after treatment with lipopolysaccharides (LPS). (**A**) The kinetics of luminol-emitted light signals after LPS treatment. Leaf disks from 3-week-old wild-type (Col-0) and *delt9* mutants were treated with 50 μg/mL LPS with the addition of luminol. Luminol-emitted light signal kinetics were recorded for 20 h. Data are the mean ± standard error (SE; n = 8). (**B**) Schematic representation of *CPR5*. Mutations in *delt9* and T-DNA insertion sites in *cpr5* are indicated. Black boxes, thin lines, and red boxes represent exons, introns, and untranslated regions (UTRs), respectively. Point mutation sites in *delt9* and T-DNA insertion sites are indicated by triangles. (**C**) CPR5-HA protein levels in two independent *35S::CPR5-HA*/*delt9* transgenic lines. Total proteins were extracted from 7-day-old seedlings and detected via immunoblotting using an α-HA antibody. Actin served as a loading control. (**D**) Luminol-emitted light signals in *35S::CPR5-HA*/*delt9* transgenic lines. Experimental conditions were the same as those used in (**A**). Total photon count for 20 h is shown. Data are means ± SE (n = 8). Letters above bars indicate significantly different values between groups (*p* ≤ 0.05; one-way analysis of variance [ANOVA]). (**E**) Transcript levels of *CPR5*. Total RNA was extracted from 4-week-old leaves. Transcript levels were detected by RT-qPCR, and *ACTIN7* was the reference gene. Asterisks indicate significant differences between *cpr5* and Col-0 (** *p* ≤ 0.01; Student’s *t*-test). (**F**) Luminol-emitted light signals in *cpr5* mutants. Experimental conditions were same as those used in (**A**). Data are means ± SE (n = 8). Asterisks indicate significant differences between *cpr5* and Col-0 (** *p* ≤ 0.01; Student’s *t*-test).

**Figure 2 antioxidants-12-00301-f002:**
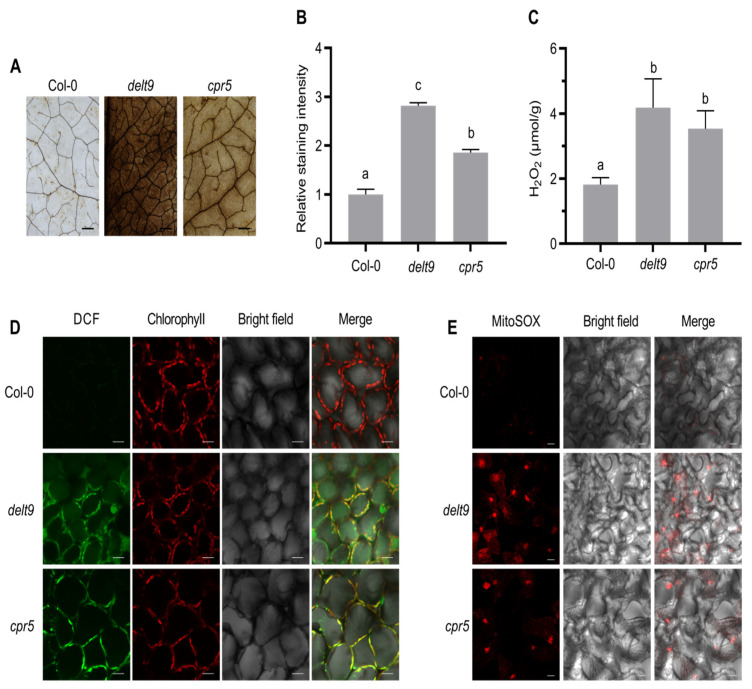
*cpr5* mutants show increased ROS accumulation. (**A**) Representative images of leaves after 3,3′-diaminobenzidine (DAB) staining. The 5th–7th rosette leaves from 4-week-old plants were detached and stained with DAB solution. Scale bars = 2 mm. (**B**) Quantification of DAB staining intensity in (**A**). Relative DAB staining intensity per unit leaf area was measured using the ImageJ software. Data are means ± SE (n = 12). Letters above bars indicate significantly different values among groups (*p* ≤ 0.05; one-way ANOVA). (**C**) Quantification of H_2_O_2_ levels by titanium sulfate colorimetry. H_2_O_2_ was extracted from the 5th–7th rosette leaves. Data are means ± SD (n = 3). Letters above bars indicate significantly different values among groups (*p* ≤ 0.05; one-way ANOVA). (**D**) ROS accumulation detected by 2,7-dichlorodihydrofluorescein diacetate (H_2_DCFDA) staining. The 5th–7th rosette leaves from 4-week-old plants were stained with 10 μM H_2_DCFDA for 20 min, and then green fluorescence was visualized by confocal microscopy. Representative images are shown. Scale bars = 20 μm. (**E**) Mitochondrial superoxide levels detected by MitoSOX Red staining. The 5th–7th rosette leaves were stripped off their lower epidermis and subjected to MitoSOX Red solution to observe their mitochondria. Scale bars = 20 μm.

**Figure 3 antioxidants-12-00301-f003:**
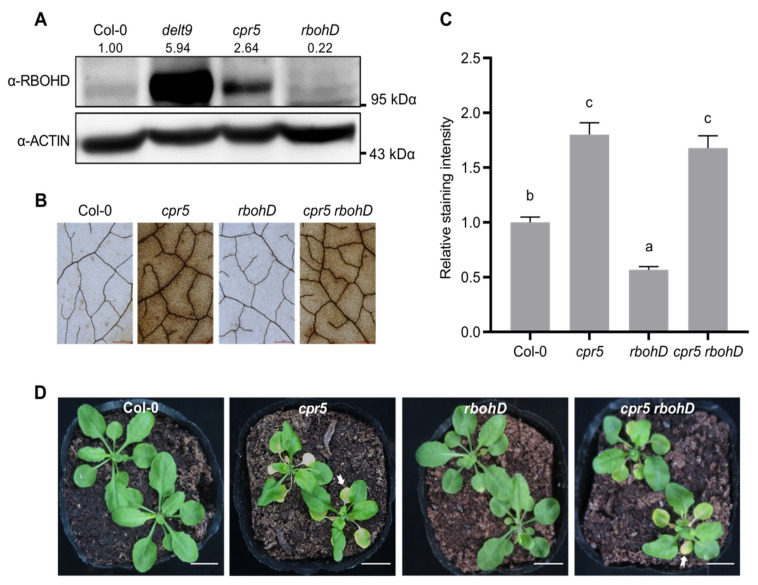
RBOHD is not responsible for excessive ROS in *cpr5* mutants. (**A**) RBOHD accumulation was higher in the *cpr5* mutants than in the wild-type (Col-0). Total proteins were extracted from 4-week-old plants and detected by immunoblot analysis with an α- RBOHD antibody as well as an α-ACTIN antibody as a loading control. Numbers at the top of the blots indicate the relative levels of RBOHD proteins normalized with the ACTIN levels. The experiment was repeated twice with similar results. (**B**) ROS accumulation was detected by 3,3ʹ-diaminobenzidine (DAB) staining. The 5th–7th rosette leaves from 4-week-old plants were detached and stained with DAB solution. ROS production was visualized in form of dark brown precipitate in detached leaves. Representative leaves are shown. Scale bars = 2 mm. (**C**) Quantification of DAB staining intensity in (**B**). Relative DAB staining intensity per unit leaf area was measured using the ImageJ software. Data are means ± SE (n = 10). Letters above bars indicate significantly different values among groups (*p* ≤ 0.05; one-way ANOVA). (**D**) Introduction of *rbohD* into the *cpr5* mutants did not prevent premature leaf senescence. The 4-week-old plants were photographed. White arrows indicate early leaf senescence. Scale bars = 1 cm.

**Figure 4 antioxidants-12-00301-f004:**
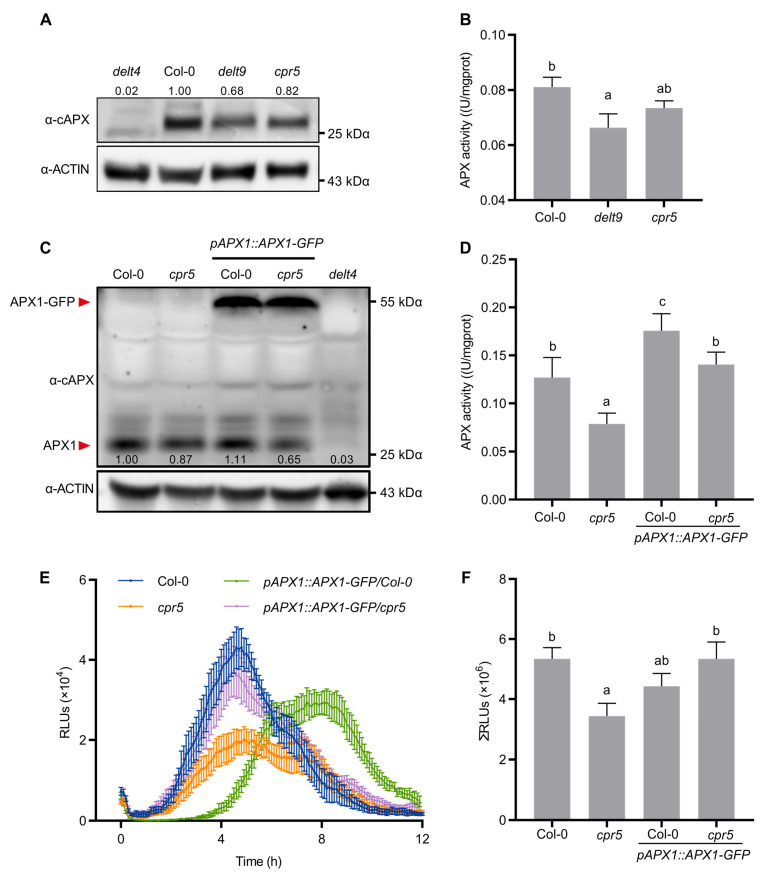
*cpr5* mutants show reduced APX1 abundance. (**A**) APX1 protein abundance. Total proteins were extracted from 4-week-old plants and detected by immunoblot analysis using an α-cAPX antibody. A null allele of *apx1* (named *delt4*) was used as a negative control. Numbers at the top of the blots indicate the relative levels of APX1 proteins normalized with the actin levels. The experiment was repeated twice with similar results. (**B**) APX activity. Crude enzymes extracted from 5th–7th rosette leaves on 4-week-old plants. APX activity was determined by ascorbate oxidation. Data are means ± SD (n = 2–3). Letters above bars indicate significantly different values among groups (*p* ≤ 0.05; one-way ANOVA). (**C**) APX1 protein levels in Col-0, *cpr5*, and transgenic plants overexpressing APX1. Total proteins were extracted from 4-week-old leaves and detected by immunoblot analysis using an α-cAPX antibody and α-actin antibody as the loading control. Numbers at the bottom of the blots indicate the relative levels of APX1 proteins normalized with the actin levels. The experiment was repeated twice with similar results. (**D**) APX activity in Col-0, *cpr5*, and transgenic plants overexpressing APX1. Data are means ± SD (n = 4). Letters above bars indicate significantly different values among groups (*p* ≤ 0.05; one-way ANOVA). (**E**) *APX1* overexpression in *cpr5* restored reduced luminescent signals. Leaf disks from 3-week-old plants treated with 50 μg/mL lipopolysaccharides (LPS). Luminescent signal kinetics recorded by chemiluminescence assay for 12 h. Data are means ± SE (n = 6–8). (**F**) Total photon count within 12 h in (**E**) is presented. Data are means ± SE (n = 8). Letters above bars indicate significantly different values among groups (*p* ≤ 0.05; one-way ANOVA).

**Figure 5 antioxidants-12-00301-f005:**
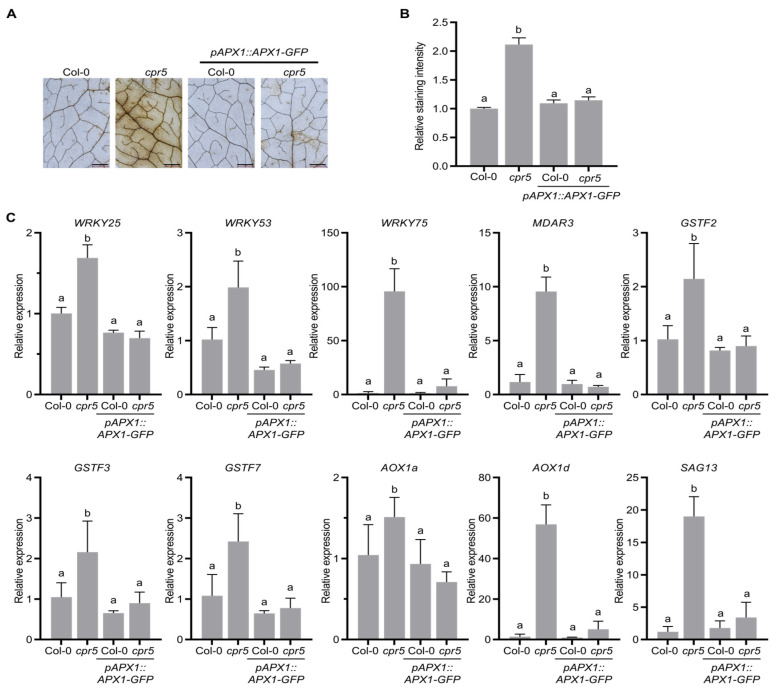
*APX1* overexpression in *cpr5* mutants removes excessive ROS. (**A**) APX1 overexpression removed excessive ROS accumulation in *cpr5* mutants. ROS accumulation was detected by DAB staining. The 5th–7th rosette leaves on 4-week-old plants were detached and stained with DAB solution. ROS production was visualized as dark brown precipitate in detached leaves. Representative leaves are shown. Scale bars = 2 mm. (**B**) Quantification of DAB staining intensity in (**A**). Relative DAB staining intensity per unit leaf area was measured using the ImageJ software. Data are means ± SE (n = 10). Letters above bars indicate significantly different values among groups (*p* ≤ 0.05; one-way ANOVA). (**C**) *APX1* overexpression in *cpr5* mutants restores elevated ROS-related gene expression. Total RNA was extracted from 4-week-old leaves. *ACTIN7* served as the reference gene. Data are the mean ± SD (n = 4). Letters above bars indicate significantly different values among groups (*p* ≤ 0.05; one-way ANOVA).

**Figure 6 antioxidants-12-00301-f006:**
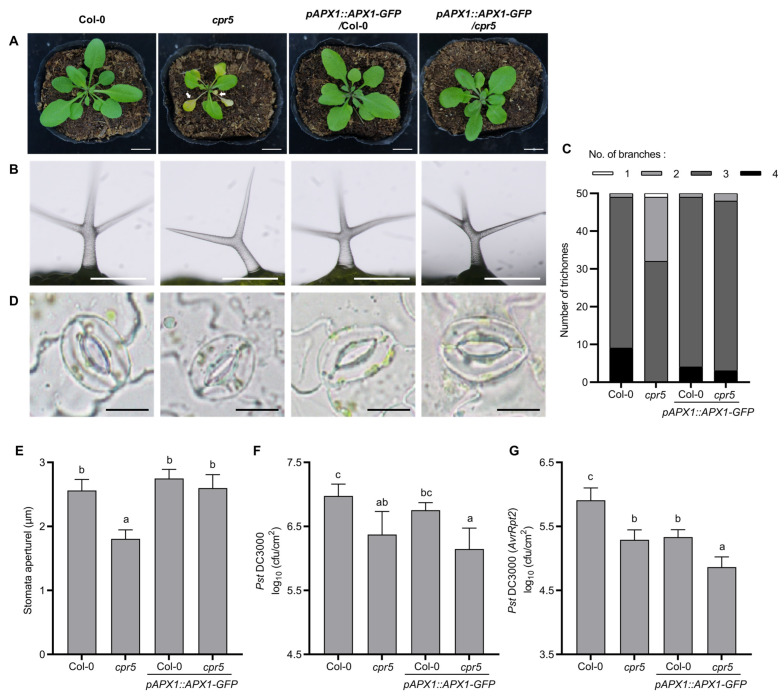
*APX1* overexpression restores the *cpr5* mutant growth phenotype. (**A**) *APX1* overexpression rescued premature leaf senescence in *cpr5* mutants. Images of 4-week-old plants are shown. White arrows indicate leaves in early senescence. Scale bars = 1 cm. (**B**,**C**) *APX1* overexpression restored reduced trichome branches in *cpr5* mutants. Trichomes from 4-week-old leaves were observed under light microscope. According to the number of branches, trichomes are divided into four classes. Representative images are present in (**B**), and number of trichomes in each class is shown in (**C**). Scale bars = 1 mm. Data are means ± SE (n = 50). (**D**,**E**) APX1 overexpression restored reduced stomatal aperture in *cpr5* mutants. Stomata of 4-week-old leaves were observed under light microscope. Representative images are present in (**D**), and the value of stomatal apertures is shown in (**E**). Scale bars = 10 μm. Data are means ± SE (n = 30). Letters above bars indicate significantly different values among groups (*p* ≤ 0.05; one-way ANOVA). (**F**,**G**) *APX1* overexpression in *cpr5* did not rescue its pathogen resistance. The 4-week-old leaves were infiltrated with *Pseudomonas syringae* pv. *tomato* (*Pst*) DC3000 (**F**) and *Pst* DC3000 (*AvrRpt2*) (**F**), and bacterial colonies were then enumerated 3 and 2 days post-inoculation, respectively. Data are means ± SD (n = 4). Letters above bars indicate significantly different values among groups (*p* ≤ 0.05; one-way ANOVA).

**Figure 7 antioxidants-12-00301-f007:**
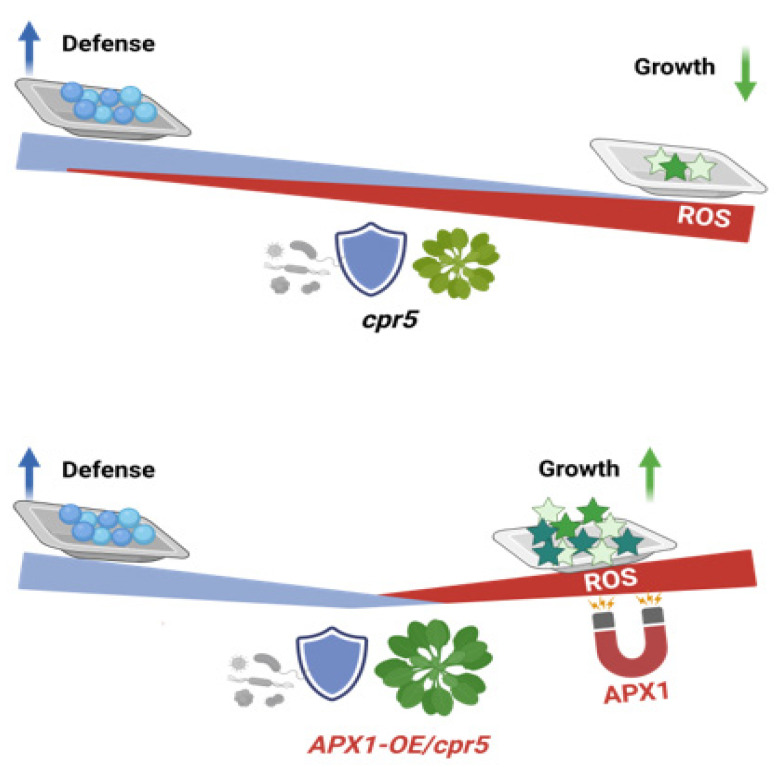
Schematic model. *APX1* overexpression in the *cpr5* mutant restored growth impairment by scavenging excess cytosolic ROS accumulation while maintaining enhanced disease resistance in Arabidopsis. Figure is created with biorender.com.

## Data Availability

Data is contained within the article and Appendix A.

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
