# Peer review of "Overexpression of an Antioxidant Enzyme APX1 in cpr5 Mutant Restores its Pleiotropic Growth Phenotype"

_antioxidants, 2023, doi:10.3390/antiox12020301_

Round 1

Reviewer 1 Report

In this manuscript, the authors found that the reduced growth and increased senescence observed in cpr5 mutants is due to an accumulation of intracellular ROS. By expressing an additional copy of the APX1 gene, they were able to reduce the levels of intracellular ROS in these mutants, leading to the restoration of normal plant growth and the reduction of enhanced senescence. Interestingly, the expression of APX1 did not affect the increased resistance to Pseudomonas, which is also a characteristic of cpr5 mutants. These findings demonstrate the importance of ROS homeostasis in plant growth and senescence, and the potential of APX1 as a target for regulating these processes.

The authors discussed the possibility that the cpr5 mutation impacts both SA-mediated immunity and intracellular ROS in a pleiotropic manner. I agree with this statement. However, while they highlight the discovery of a new allele of cpr5, it is worth noting that this allele is not significantly different from previous knockouts of the CRP5 gene. Instead, the major finding of this paper is the demonstration of the cpr5 mutation's effect on SA-mediated immunity and intracellular ROS. This information is important for understanding the role of the CRP5 gene in plant defense and could potentially lead to new strategies for improving plant immunity. Therefore, the focus of the manuscript should be on this key finding rather than the specific details of the new allele.

The writing contains many small inaccuracies but in general clear and grammatically well-written. The following are some minor errors pointed and suggestions:

Page 1 line 11 “a new allele of cpr5 mutant” wrong phrase. It should be “a new allele of CPR5 gene”.

Page 2 line 55 citation is out of style.

Page 2 line 56 remove “et al” in front of citation [21].

Page 2 lines 60-79 Irrelevant for this introduction. You can move this paragraph into Methods or delete completely.

Page 2 line 80 It is not necessary to provide details of previously published work; a citation is sufficient. Therefore, I suggest that the following text be written: "In this study, it was established that the previously found delt9 mutation [33, 39] was a novel allele of the CPR5 gene."

Page 14 line 487 citation is out of style.

Reviewer 2 Report

In this work the authors demonstrate restoration of growth retardation phenotype in the cpr5 mutant through over-expression of APX1 and thereby reducing cellular ROS content. Over-expression of APX1 did not compromise the disease resistance phenotype in the cpr5 mutant against Pseudomonas syringae pv. tomato (Pst) strains. The manuscript is well written, and the results support the conclusions. I have some minor comments to improve the current content.

Line 22: Can results obtained using Pseudomonas syringae pv. tomato (Pst) be generalized for overall plant disease resistance? Did the authors check resistance of the cpr5 mutant against other plant pathogens such as fungi, viruses etc.?

Line 126: 2‐ΔΔCt method: The authors should provide appropriate citation.

Line 128: Is there an appropriate citation for this section (Protein extraction and immunoblot analysis)?

Line 143: The authors should briefly mention the steps in addition to the citation used for this method (Measurement of APX activity).

Line 416: ‘Figure 6. APX1 overexpression suppresses (restores?)…..
